# Work Methods for Nursing Care Delivery

**DOI:** 10.3390/ijerph18042088

**Published:** 2021-02-21

**Authors:** Pedro Parreira, Paulo Santos-Costa, Manoel Neri, António Marques, Paulo Queirós, Anabela Salgueiro-Oliveira

**Affiliations:** 1Enfermagem, Escola Superior de Enfermagem de Coimbra, Unidade de Investigação em Ciências da Saúde, 3004-011 Coimbra, Portugal; pauloqueiros@esenfc.pt (P.Q.); anabela@esenfc.pt (A.S.-O.); 2Conselho Federal de Enfermagem (COFEN), Brasília 70736-550, Brazil; manoel.carlos@cofen.gov.br; 3Centro Hospitalar e Universitário de Coimbra, 3004-561 Coimbra, Portugal; amarques@chuc.min-saude.pt

**Keywords:** nursing care management, care delivery methods, nursing services, organization and administration, nursing process

## Abstract

This article analyzes the work methods based on care design, identification of needs, care organization, planning, delivery, evaluation, continuity, safety, and complexity of care, and discharge preparation. It describes the diagnosis of the situation, goal setting, strategy selection, implementation, and outcome evaluation that contribute to adopting a given work conception and/or method for nursing care delivery. Later, the concepts underlying the several methods—management theories and theoretical nursing concepts—are presented, with reference to relevant authors. The process of analysis and selection of the method is explained, highlighting the importance of diagnosis of the situation, goal setting, strategy selection, implementation, and outcome evaluation. The importance of various elements is highlighted, such as structural aspects, nature of care, target population, resources, and philosophy of the institution, which may condition the adoption of a method. The importance of care conceptualization is also underlined. The work methods are presented with a description of the key characteristics, advantages, and disadvantages of the task-oriented method (functional nursing) and patient-centered methods: individual, team nursing, and primary nursing. A critical and comparative analysis of the methods is then performed, alluding to the combination of person-centered methods.

## 1. Introduction

In any healthcare institution, the goals of safe and successful care delivery include the pursuit of excellence in quality and safety at the lowest cost while improving patient and family satisfaction outcomes [1]. However, to achieve these objectives, the organization must implement a care delivery model that fits its vision and mission and combines the human and material resources available. Nurse managers are responsible for these tasks, including the selection of nursing care organization methodologies and the creation of conditions for their implementation, given their influence in professional practice and the decision-making processes of healthcare institutions [1]. 

The methods are based on administration and nursing theories and refer to how work is conceptually structured, organized, and allocated to nurses. Organizational methods define how nurses organize and distribute work with the purpose of providing efficient care in an environment where safety issues are a major concern. Therefore, they should not be seen simply as an assignment of activities but also as a way in which nurses choose and adopt a philosophy of care. According to Fowler and colleagues [2], work methods represent the structural and contextual dimensions of nursing practice, determining how nurses organize their work, communicate, interact with team members and other health professionals, and make clinical decisions. In this way, care is delivered based on specific communication and coordination patterns. From a management point of view, the methods adopted for care organization should reflect the conceptual framework, the perspectives, and the theories which the manager will use to promote the delivery of safe and efficient nursing care. Thus, the work methods are translated in the adoption of dynamic principles that promote a strong interaction in interpersonal relationships, taking into account the patient’s level of dependency, as well as the quality and safety of care, with the patient’s experience and satisfaction in mind [3].

The selection of a nursing work method reflects the type of care conceptualization, organization, and delivery in a given context. The work methods translate a perspective, a philosophy of care, a way of thinking about and organizing care delivery by a nursing team. They also reflect the nurses’ beliefs, conceptions, and principles that sustain care design, guiding them to obtain results [1]. Thus, the delivery of nursing care is expected to comply with the criteria that support the method adopted. As an example, if the nursing team believes that each patient has the right to high-quality care, all nurses are expected to incorporate this vision into their practice, translated into a more demanding method in terms of quality and safety in care delivery and excellent professional performance.

## 2. Management Theories

Management theories and nursing theories support and give form to the work methods [4]. Historically, it is possible to identify the paradigmatic milestones in management and nursing theories that have influenced the way health organizations and the role of nursing and nursing care are conceived (Figure 1). In this line of thought, which reflects the disruptive conceptions that characterized this last century (e.g., shift from task to person-centered care, complexity as opposed to linear thinking), it became necessary to reconfigure work methods for nursing care delivery with a view to maximizing the available resources and the efficiency, quality, and safety of care.

Management theories have been intensively developed over the last decades [5]. It is possible to historically demarcate the classical approaches (the bureaucratic or scientific management theory), the neoclassical/institutional approaches (organization as a social arena or limitation of rationality), the systemic approaches (general systems theory or the organization as an open system), the contingency approaches (management and environment or management and innovation), the interactionist approaches (symbolic interactionism), and the postmodern approaches (epistemological postmodernism).

The concept of organization, including its identity, limits, or roles, has undergone profound changes, impacting how its stakeholders are organized. These transformations have also influenced the methods of organization, planning, and control of the nursing care process. From a purely historical viewpoint, the work methods evolved according to economic issues in the 1930s, political issues in the 1940s and 1950s (World War II and immediate post-war), and emerging social considerations such as the humanist aspect in the 1960s [6]. However, in recent years, attempts have been made to revitalize traditional approaches, combining characteristics of the four main methods [7]. 

In 1909, Frederick Taylor argued in his book *Principles of Scientific Management* that the work of all organizations should be divided into specific sections, as in an assembly line, with the supervision of a single section manager [5]. By observing a worker’s physical movements while performing a given task, Taylor introduced the concept of worker-machine in the organizations, emphasizing the need for each individual to specialize in a specific activity. In the Bureaucracy Theory, Max Weber (1864–1920) advocated a clear distinction between areas of competence, guided by laws and written regulations. There was a hierarchy in organizational roles and a strong sense of supervision by levels [5]. On the other hand, Henri Fayol (1841–1925) identified fourteen management principles, such as the division of labor (specialization leads to greater efficiency), authority and discipline, centralization, unity of command (avoiding confusing/conflicting instructions), and unity of direction (individuals with the same goal should work under the direction of a manager, using a single action/intervention plan). 

Therefore, in this period, there was a need for individual workers to perform single specialized tasks while complying with well-defined standards and regulations, under the responsibility of a hierarchical supervisor. In this way, a parallel can be drawn between the management theories in force at the time and how nurses were organized in the institutions, with a predominance of the functional method of nursing care delivery (task-centered), that will be addressed later in this chapter. Until then, the worker was perceived as an extension of the machine. However, at the end of the 1920s and the beginning of the 1930s, the Behavioral Theories emerged in organizational contexts, focusing on group dynamics, complex worker motivations, and the importance of leadership as a source of inspiration and collective vision. For example, Elton Mayo (1880–1949) focused his attention on the importance of groups in organizational contexts. Mayo found that workers thought and acted not as individuals but as a group, usually sacrificing their interests in the face of group pressure [5]. It was within this framework that the importance of formal and informal group dynamics in individual productivity began to be understood. Mayo concluded that managers should focus on teamwork rather than on individuals without neglecting each worker’s needs and sources of satisfaction. In turn, Mary Parker Follett (1868–1933) believed that management is a dynamic, not static, process that is done “through people” [5]. Follet believed that the best decisions in a unit/department are made by the employees closest to the action, which is the reason why they should be involved in the decision-making processes.

From this point of view, we witnessed the implementation of organizational assumptions centered on interrelationships, group dynamics, and individual motivations, which were reflected in how the workers, previously disconnected from the whole, were coordinated and integrated into the management processes [8]. This behavioral theory strongly influenced the reformulation of health organizations, leading to the emergence and implementation of patient-centered methods for nursing care organization, planning, and delivery, especially the individual method and the team nursing method. 

However, in the 1970s and 1980s, new forms of organizational management not only reinforced the implementation of the individual and team nursing methods, which still exist today in international health services, but also fostered the emergence of a new patient-centered care method: primary nursing. In 1970, from an interactionist perspective, David Silverman, in his work *The Theory of Organizations*, stated that a set of non-random interrelationships characterizes organizations. In 1969, Karl Weick, in his work *The Social Psychology of Organizing*, viewed organizations as sensemaking systems that incessantly (re-)create their own conception. Later, in 1990, Stewart Clegg, in his work *Modern Organizations*, stated that organizations are composed of (collective or individual) agents who act under a subjective rationality [9] and are human fabrications, as it is also argued by other relevant postmodern authors.

## 3. Nursing Theories and Concepts

The evolution of the nursing care delivery methods can be understood by analyzing the state-of-the-art evolution of nursing theory. In this context, the contributions of Meleis [10], Kim [11], Fisher and Tronto [12], Tronto [13], Tronto and Kholen [14] are important, namely for their synthesis and capacity to explain the key characteristics of nursing science, the assumptions of nursing as a science, and the characteristics of the plural dimension of care.

According to Meleis [10], the key characteristics that determine nursing science are summarized in four points: (i) the nature of nursing science as a human science; (ii) nursing as a practice-oriented discipline; (iii) nursing as a caring discipline, with the development of a caring relationship between nurses and patients; and (iv) nursing as a health-oriented discipline and from a perspective of well-being. According to Meleis, nursing is a human science, a practice-oriented discipline. The praxeological dimension of nursing exists and is justified by the practical actions (nursing care), posing the question of the nature of their knowledge, that is, it leads to the clarification of the epistemological affiliation of the knowledge that nurses create and recreate when caring for people. 

This epistemological duality of a positivist epistemology is based on a rationality (a form of knowledge creation), especially due to experimentation in which the conclusions are put into action (care) through the application mediated by technique and technology. On the other hand, due to an epistemology of practice, which results especially in reflection in action, the knowledge produced is implemented by translation rather than by application. Practical research and theorization are mixed in a productive and effective whirlwind, finding solutions to concrete questions, appropriate to the moment and circumstances, in a process that some call “hermeneutic spiral”.

We believe that the definition of nursing science as a practice-oriented human science is much more aligned with the epistemology of practice, bringing about advantages for professional, disciplinary, and academic autonomy. It should also be noted that, in the synthesis by Meleis [10], nursing is a discipline of caring. The author attributed a central role to caring, which exists in the form of a relationship focused on health and well-being. It is no longer the narrow purpose of preventing and fighting illness but the wider human ambition of living with health and well-being, from a salutogenic perspective. The transitions theory offers a structure that guides the nurses’ professional actions towards helping people experience multiple transitions in a healthy way. Examples of transitions that generate vulnerability throughout the life cycle include illness, developmental transitions such as pregnancy, birth, parenthood, adolescence, menopause, aging, and death, and social and cultural transitions such as migration, retirement, and assuming the role of family caregiver [15]. The changes in individuals and families that occur during these transitions are so different that nurses need a tool that allows them to understand each individual’s experience of a transition. The transition is not an event, but the reorientation and self-definition that people go through to incorporate change into their lives (idem).

If these characteristics clarify our disciplinary framework, the contributions of Kim [11] are important for deepening its understanding. This author presents the six assumptions of nursing as a science: (1) human beings are complex; (2) one cannot know human beings as a whole; (3) nursing practice requires mutuality; (4) nursing practice is founded upon normative, moral, and aesthetic principles that go beyond scientific knowledge; (5) nursing knowledge is complementary and inclusive rather than competitive and exclusive; and (6) nursing knowledge is the knowledge of synthesis that can be revealed only by accessing the practice.

Although it seems obvious, the assumption that human beings are complex beings is essential to understanding the unpredictability of the contexts, circumstances, and human responses to the problems related to the health, disease, and well-being of our patients. Nothing is stable, and the most unstable elements are undoubtedly the human beings in their various physiological, social, and psychological dimensions and responses to problems. They expect help and strong support from their caregivers. Considering a rationalist epistemology instead of an epistemology of practice based on reflection and action implies, once again, the assumption of nursing as a practice-oriented human science, as noted by Meleis [10,15]. The paradigm of complexity, as opposed to linear thinking, has a structuring effect on nursing work methods.

Kim’s second assumption [11] that one cannot know human beings as a whole diverges from the assumption of holism and the uniqueness of each human being in itself and in each circumstance. According to Morin [16], when considering holism as the same scheme of thought as reductionism, although at the opposite pole, it is necessary to change the scheme of thought to the paradigm of complexity. Thus, another approach emerges with the dialogical, recursive, non-linear thinking that considers the human being as a self-eco-organized being. Kim [11] also refers to mutuality and that nursing knowledge is complementary and inclusive, rather than competitive and exclusive. The importance of these statements translates into the demonstrated advantages of interdisciplinary and transdisciplinary work and the gains in effectiveness and efficiency when there is planning, consistency, and solidarity work within a team to achieve previously agreed goals.

Kim’s [11] reference to nursing practice is based on normative, moral, and esthetic principles that go beyond scientific knowledge, leading us to the nursing knowledge standards that were disciplinarily introduced in the synthesis by Barbara Carper [17] and developed by so many other nursing thinkers. Thus, it is relevant to affirm that scientific knowledge is not enough for caring. Caring goes much beyond scientific (empirical) knowledge; it benefits from and needs knowledge and skills from areas such as ethics, personal knowledge, aesthetics (understood as sensitivity, intuition, and technique), socio-political knowledge [18,19,20], as well as environmental and emancipatory knowledge [21]. Finally, in the sixth assumption, Kim [11] states that nursing knowledge is ultimately the knowledge of synthesis that is revealed only by accessing the practice. In doing so, she tells us that it is not only scientific knowledge but a synthesis of knowledge of various dimensions, with the possibility for practical-reflective rationality in nursing knowledge.

Assuming that the dimension of caring is what nurses do and how they identify themselves, it is important to analyze the most recent literature about this concept. Fisher and Tronto [12], Tronto [13], and Tronto and Kholen [14] refer to caring as a complex process involving multiple steps that initially consisted of four phases to which a fifth one was later added: (i) to recognize a need for care; (ii) to take responsibility; (iii) to provide care, to help, to carry out the activity; (iv) to receive a response from the impact on the other; and (v) to create conditions for it to happen. This plural dimension of the purpose with which nurses identify themselves determines the entire structure of their work (of caring) on their own initiative or within a team, in the definition of solidarity and solidarity action, forcing the implementation of scientific work methodologies in the organizations.

The implementation of a work method will have to consider the onto-epistemological conceptions adopted in care delivery and the specificity of the complexity of the health context under analysis.

## 4. Analyzing and Selecting a Work Method for Nursing Care Delivery

The principles of health planning allow making decisions based on a process that consists of diagnosis, goal setting, strategy selection, implementation, and outcome evaluation (Figure 2).

Thus, inherent to the organization of care, the following steps will have to be completed: In the first phase (analysis/diagnosis of the situation), it will be important to take into account the following aspects: (i) the physical structure, the architecture, the dimensions, and the age of construction (modern, old, adequate, inadequate) of the organization can influence the adoption of a work method (it is easily understood that centralized physical structures facilitate patient surveillance, being more consistent with the adoption of work methods of higher quality care); (ii) the institution’s characteristics regarding the type of management, its philosophy, objectives, and culture, that is, organizations that place great emphasis on their mission of providing high quality care are supposed to adopt more demanding work methods in care delivery; (iii) the nature of care, since the units providing differentiated/specialized care or primary/general care may also dictate differences in the adoption of different methods according to their specificity; (iv) the target population and its characteristics are particularly important, with emphasis on the uniqueness of the person, his/her health condition and personal, social, and family life, with an impact on the care required because factors such as the population’s age, the patients’ degree of dependency, the intensity and severity of the transitions experienced and consequent adaptation, the number of patients, and the socioeconomic level also influence the adoption of an adequate work method.

In the second phase (goal setting), the conceptualization adopted in care delivery, the management philosophy for nursing care delivery, and the availability of resources should be considered. The institution’s conceptualization and philosophy are of utmost importance because they require a specific structure and determine the adoption of principles and criteria that feed into the model and philosophy adopted. A clear and patient-centered philosophy will be an essential pillar for selecting the work method to be adopted by the institution. This method emerges from the conceptual framework for care delivery adopted at the institution, whose principles should be reflected in its mission. 

The resources involve the technical, human, and material resources available. Staffing, associated with the level of competencies in the budgetary/financial dimension, should be considered. However, it should be noted that the adoption of task-oriented methods (low-quality methods that are being put aside) rather than person-centered methods impact the patient, the professional, and the organization, which, although they may create an apparent economy because they require low staffing, are quickly overtaken by the tremendous impact of poor quality care, often leading to high rates of avoidable errors and accidents due to related adverse events.

Adopting these strategies seems to promote an idea of saving in material and human resources, assuming a priori lower-quality care. This strategy culminates in higher healthcare costs.

The third phase (strategy selection) intends to choose an appropriate course of action, taking into consideration the various constraints, that is, it aims to identify the care delivery method that best suits the specific situation. To this end, after identifying the advantages and constraints of each proposal, the most promising strategies should be selected.

In the fourth phase (implementation), the nurse manager should prepare the team to promote their adherence to the selected strategies, both in training and motivation. This phase should consider the culture and environment of the unit (or the organization as a whole) so that the selected strategies reflect the organizational values and identity. 

The fifth phase (outcome evaluation) intends to evaluate the quality of care and analyze deviations from the objectives set out, enabling corrections in the method adopted, including the definition of new guidelines for care delivery. In this phase, the adopted method should be evaluated, and the difficulties/limitations/constraints that arose should be analyzed.

### 4.1. Methods of Organization and Delivery of Nursing Care

Traditionally, four dominant methods are mentioned in the literature on the organization of nursing care delivery: functional nursing, individual, team nursing, and primary nursing [4]. The work methods are conceptually organized into two major groups: the task-oriented method and the person-centered methods (Table 1). These methods are internationally recognized and commonly applied in clinical settings to support the organization in nursing care delivery, reflecting the social values, management ideologies, and resources of a team or organization [6]. 

#### 4.1.1. Functional Nursing Method (Task-Oriented) 

The functional nursing method became popular during World War II, given the need for nurses to care for many wounded people in hospital settings [22]. The delivery of nursing care was based on the distribution of standardized tasks by the nurses, who achieved proficiency through the systematic repetition of techniques (such as intravenous drug administration and vital signs monitoring). However, with the end of the war and the sudden increase in birth rates in the following years, this method of care delivery persisted and is still used today in specific clinical contexts [22], which is considered inappropriate given the associated risks for the quality of care delivery. 

Functional nursing, also known as task nursing, focuses on the distribution of work based on the performance of tasks and procedures, where the target of the action is not the patient but rather the task [23]. The work is thus broken down into tasks performed by different professionals, from a mechanistic perspective [1]. The adoption of this method in care organization is based on Taylor’s principles of the industrial revolution, promoting the maximization of the task in a routine and mechanistic logic. As shown in Figure 3, this care delivery model is characterized by a lack of coordination between the parts, represented by “piecemeal” interventions in task-oriented care delivery, of an interrelated whole (non-holistic care to the patient). However, a significant number of health managers and administrators believe that “functional” nursing is an economically efficient care delivery method [22]. 

This method of classical inspiration in the industrial revolution was used when human resources were scarce because it required a smaller number of professionals [1] or when it was intended to develop manual dexterity/improve a technique to obtain a better performance. The functional nursing method reduces communication between health team members. The performance of tasks is assumed as the primary purpose in this method, where each nurse “routinizes” the provision of care instead of adopting procedures to provide personalized care to meet each patient’s needs [4]. This fragmented approach sees the patient as a “place” where nursing care is provided [1], without significant advantages for the patient. A single advantage for nurses is the increase in manual dexterity when approached as a technique and not as a method. For the organization, the advantages include its immediate efficiency that is translated into an apparent improvement in productivity since it requires fewer nurses, despite the costs derived from the disastrous consequences on quality of care due to safety failures with accident rates and avoidable adverse events [6].

However, there are several disadvantages. For patients, it does not allow for personalized care, which dehumanizes them, leading to poor quality care and major flaws in patient safety when compared to other methods; it causes more complaining from patients, who are divided by tasks and, consequently, by different professionals, promoting unaccountability; it does not allow the delivery of comprehensive care; it damages the nurse-patient relationship because the patient is not familiar with “his/her” nurse and promotes patient insecurity [6].

For nurses, it does not allow the application of the nursing process, leading to major difficulties in the identification of patient needs and poor records; it does not promote the continuity of care; it leads to some activities being “forgotten” due to lack of planning; the nurse does not have an overall view of “his/her” patient; it hinders the assessment of care; it increases the risk of healthcare-associated infections (HAIs); it hinders the interaction and the interpersonal relationships between health professionals; it creates poor team spirit and lack of motivation by “routinizing” tasks that are repeatedly performed by the same nurses (Taylorism). Although this method does not currently have a framework in nursing theories, it still seems to be alluded to and practiced based on the classical theories of the industrial revolution. For the organizations, this method results in poor quality and dissatisfied patients.

#### 4.1.2. Person-Centered Care

In person-centered methods, care is delivered according to the scientific method, starting with the identification of the nursing care needs, the definition of priorities, planning, implementation, and evaluation of interventions, providing individualized and personalized care to the patient as an interconnected and integrated whole. It is part of the humanistic perspectives, valuing the relationship and the concern for the other, where the whole is more than the sum of its parts. Thus, it comes closer to the systemic-contingency perspectives. 

##### Individual Method

Known as the case method or total patient care approach, it corresponds to a situation where a single nurse assumes full responsibility for delivering care to a group of patients during a shift [24]. Although care is not fragmented, its coordination does not prevail between shifts, and changes may occur in the established nursing care plan [1]. In this method, the overall organization of care to meet the needs identified by the nurse depends on the nurse’s view of his/her role as a professional and may prioritize the patient or the performance of tasks (Figure 4). In addition, and because the individual method limits the nurses’ action during a shift and the patient(s) to which he/she is allocated, outcome evaluation is based only on circumstantial objectives [22]. The coordination of the care delivered to all patients in the unit is under the responsibility of a single nurse, usually the head nurse, who supervises and evaluates the delivery of nursing care and makes the most significant decisions throughout the process. However, care delivery in that shift is delegated to the nurse allocated to that shift. 

This method has the following advantages for patients: the individualization of care, with satisfaction of their needs; it promotes the nurse-patient relationship; the patient can identify the nurse who provides care in a given shift, resulting in a close, humanized, and personalized care, which also reinforces the confidence in the nurse and the patient’s safety.

For nurses, it promotes the application of the nursing process and the consequent identification and satisfaction of patient needs, with a better ability to make decisions while considering the biopsychosocial totality of human beings; it allows the continuity of care and leads to a reduction of the number of errors and/or omissions, increasing quality; the patient’s deeper knowledge of the nurse leads to greater satisfaction and less effort by establishing an intentional therapeutic relationship; it leads to the development of the nurses’ individual skills and abilities; it promotes greater autonomy in care delivery, responsibility, creativity, knowledge updating, and the evaluation of the quality of care delivery. For the organizations, it translates mainly into higher quality in care delivery [6].

This method also has some disadvantages/weaknesses. For patients, as each nurse will provide care to different patients in each shift, there is no nurse or figure of reference to whom he can talk to during the hospital stay as “his/her” primary nurse. Due to differences in the individual skills and knowledge of the nurses who provide care, asymmetry can be created in care delivery, leading to heterogeneity in the several shifts related to different levels of care being delivered. For nurses, it has some potential for emotional involvement with the patients and requires a higher level of proficiency from all nurses. For organizations, it requires more staff than the task-oriented method.

##### Team Nursing Method

This method emerged in the 1950s as a response to low nurse staffing and widespread dissatisfaction among the professionals and patients with the functional nursing method [24]. It was created to take advantage of the mix of skills of the several members of the nursing teams. This method implements a philosophy where a leader leads a group of people. Thus, it is not considered a procedure. In this philosophy, all team members are familiar with the patient’s needs and/or problems, contributing in a particular way to his/her well-being [22]. Thus, the team nursing method assumes that the care provided will be of higher quality, efficacy, and safety when planned and provided as a team, due to the individual contributions of each nurse [7]. Nurses are divided into teams and guided and coordinated by leaders, maximizing the group’s capabilities and the individual qualifications and skills of each nurse [1]. Thus, each team is responsible for the full delivery of care to the patients who are under its responsibility (Figure 5).

This method is based on two fundamental pillars: (i) leadership in the planning and evaluation of care delivery to each patient; and (ii) effective communication to ensure continuity of care [7]. According to Kron and Gray [23], it rests on the principle that all patients can receive better care, provided under the leadership of a nurse who is responsible for: (i) assessing each patient and determining the appropriate nursing intervention; (ii) coordinating planning and intervention; (iii) keeping care plans up to date; and (iv) ensuring that the records of care delivery are made.

In the literature, there are some misconceptions about the team nursing method, which, according to Kron and Gray [23], are unjustified. According to the authors, the team nursing method does not correspond to the distribution on a functional basis to the various team members or the division of units. Further pointed out as a fragility, it does not refer to the division of responsibilities and patients equally among team members, that is, it does not consist only of cooperation or teamwork. The team nursing method aims to decentralize responsibilities, fighting against the traditional notion of nursing leadership and management concentrated on a single individual. Thus, each nurse recognizes the patients assigned to their team, contributing individually to the satisfaction of their needs. This reality facilitates the consensus among nurses regarding the nursing care provided in the units, laying the foundation for the creation of a school of leaders. 

This method has the following advantages/strengths for patients: it promotes a feeling of security as care needs are met by a group of nurses with different levels of skills; it leads to effectiveness in care delivery; and it promotes close and personalized care, increasing patient satisfaction. For nurses, it values and increases communication, teamwork, and leadership; it ensures support from the most experienced and competent (experts) nurses to recent graduates, who are less competent and less experienced; it allows for the development of knowledge and skills through effective work; and it ensures the identification of training needs. For the institutions, it promotes leadership skills, taking advantage of the competencies and skills of all team members; it increases satisfaction; and it facilitates the implementation of new methodologies within the team.

However, its application also has limitations. The patient cannot identify a primary nurse who cares for him/her, which hinders referral. It does not guarantee care delivery in another shift by the same team. For nurses, it can fall into the task method because it requires more effort, dynamism, and quality in interpersonal relationships, and difficulties may arise in interaction and collaboration. Nevertheless, if there is a conceptual awareness of the philosophy underlying this method, this problem will not arise. It may also lead to an unfair distribution or omission of some activities that ought to be performed and even incompetence by some members. It also requires greater knowledge of several patients. For the institutions, it also requires a higher number of nurses than the functional nursing method and the individual method.

##### Primary Nursing Method

This work method was first called primary nursing, and it was developed in 1968 under the direction of Marie Manthey at the University of Minnesota Hospitals, USA [25]. The primary nursing method is based on the idea that a nurse is responsible for planning, delivering, and evaluating the care of one or more patients from the moment of admission to discharge [22]. 

In order to allow continuity of care, each primary nurse is assisted by associate nurses. The delegation of care to associate nurses occurs whenever the primary nurse is not present. Thus, nurses may be simultaneously primary nurses for some patients and associate nurses of other nurses (Figure 6). However, ultimately, the primary nurse will always be responsible for the coordination of the clinical decisions and supervision during the hospital stay [1].

This method has several advantages. For the patient, it allows them, better than any other method, to identify who will be the primary nurse during the hospitalization, ensuring continuity of care as no other method [26], reducing the number of nurses issuing nursing prescriptions/interventions, and reducing the likelihood of errors. The primary nursing method promotes the relationship between nurses and patients, allowing personalized care and increased satisfaction [27]. For the nurse, it promotes the relationship with the patient, maximizes the identification of needs (due to the deep knowledge of the patient’s status), and the capacity for prescription and evaluation, and allows the identification of the outcomes of their work in terms of health gains, such as a reduction in the number of infections associated with urinary, central, and peripheral catheters [28]. It also facilitates the communication between nurse/nurse, nurse/patient, nurse/family, nurse/doctor, and nurse/other health professional; it facilitates the training of patients and/or informal caregivers, and promotes a more effective and efficient discharge planning [29]. Furthermore, it gives greater autonomy to nurses, personalizes responsibility, values nurses’ autonomous interventions, and promotes creativity [30]. For the institution, this method ensures a higher quality of care, increases patient/family satisfaction, and facilitates the internal mobility of staff and the maximization of their performance [29].

It also has some disadvantages for the patient, such as the discontinuity in the quality of care that depends on the nurses’ qualifications. In turn, it limits the differentiation and performance of associate nurses, carries a risk of emotional involvement with the patient, and can increase stress [31]. Given that not all nurses have differentiated training, knowledge, and decision-making capacity, some nurses will not be able to assume the responsibility of coordination [32,33]. For the institution, the implementation of this method requires more staff, raises questions regarding the equity of care, requires greater preparation of associate nurses, and requires greater investment in team staffing.

However, according to Kusk and Groenkjaer [25], although the primary nursing method was widely disseminated more than 50 years ago, only a few studies have been conducted to identify its advantages. In fact, according to the authors, the evidence that supports the superiority of this method over those previously presented “is mainly based on anecdotal or descriptive studies” rather than on quantitative data derived from robust research studies, with genuine control units/institutions, outcomes of interest, and appropriate measurement instruments [25], despite the results that the research may show.

#### 4.1.3. Critical and Comparative Analysis of Nursing Work Methods

There are four nursing work methods identified: functional nursing, individual, team nursing, and primary nursing. Although conceptually, the task-oriented work methods do not show similarities with the person-centered methods, it is still possible to represent their key aspects in a single table, distinguishing them in terms of complexity of care delivery (Figure 7).

Thus, the primary nursing method seems to be the one with the most developed key aspects, contrasting with the functional nursing method. In addition, it is the method that most promotes planning, autonomy, responsibility, decision-making skills, and continuity of nursing care. 

Sellick and collaborators [34] compared the functional nursing method and the primary nursing method in two internal medicine units and identified statistically significant differences between the levels of satisfaction of inpatients. In the units under the primary nursing method, the patients reported that the nurses had a more global view of their needs and, consequently, were more concerned with care, communicated more with them and their families, and planned their discharge in a timely and gradual way. From the nurses’ perspective, the primary nursing method increased their satisfaction, enabled them to use all their skills, and made them feel more fulfilled by the profession [34].

However, Fernandez and collaborators [35] found contrary results. The adoption of the team nursing method showed a higher level of documentation of care and earlier discharge planning when compared to the primary nursing method. This fact indicates that implementing the primary nursing method in a clinical context is not always a success [1] because it can be adopted in an inconsistent and unstructured manner, not taking into account the level of competence of the primary element. Huber [1] tells us about this barrier to more inexperienced nurses, which may culminate in burnout for nurses who are fully responsible for delivering care to a patient without proper preparation. There are limitations in the implementation of this method related to budget and pressure for shorter hospitalizations despite the complexity of care [36]. 

A systematic review of the literature conducted by King and collaborators [7] sought to determine the effectiveness of the team nursing method when compared to the individual method, considering staff well-being (assessed through their satisfaction, absenteeism, stress, and burnout). Although differences were found in specific subdomains of satisfaction, the care organization method in force was not significantly different for nurses. Given their importance for nurses, the authors found that the available human resources, the stability in staffing, the size of the units, the professionals’ skill mix and experience, as well as the ethos of care, should be considered when implementing a care delivery and organization method. However, this process should be based on effective leadership, creating a supportive environment for all professionals [7].

In a systematic literature review with 2000 participants, Fernandez and collaborators [35] found that the work methods influenced not only the care provided to the patient and family but also the dynamics of the clinical context and existing professional interrelations. Concerning the delivery of care to patients, the authors did not find consistent results regarding the incidence of falls in services under the teamwork method. However, for the same method, two studies showed that its adoption in the clinical context is statistically associated with lower levels of pain reported by patients and lower incidence of medication errors and adverse outcomes. They also found that the delivery of care using a “hybrid” model, that is, a combination of characteristics of the individual method and the team nursing method, resulted in improved quality of care, especially regarding a reduction in restraint use, but without any impact on the incidence of pressure areas or healthcare-associated infections. 

Thus, it is up to nurse managers to recognize the needs and strengths of their nursing team based on their interrelationship with the extended health team and the patients, contextualizing them in a specific sociocultural, geographical, economic, and temporal background. In this line of thought, the discussion of the work method to be used in each clinical context should focus on its dynamics, organization, level of complexity, and required conceptual framework.

One could ask “What is the best response to our patients’ nursing care needs?”. In patient-centered organizations, with a high focus on the promotion of quality of care, more than choosing a crystal-clear model, it is important to ensure the satisfaction of the patient’s nursing needs, preferably with gains for all those involved. In our opinion, supported by the theoretical-conceptual framework of management and nursing and the available studies, this can be achieved through a combination of methods, that is, hybrid methods if they are patient-centered. 

Several authors believe that the primary nursing method is the one that best answers the issues of humanization and quality of nursing interventions, with the patient and his/her family at the center of care [1,37,38]. According to Wessel and Manthey [38], this method is the one that best supports the nurses’ professional practice, focusing on the nurse/patient relationship, facilitating the participation and involvement of the family in care delivery, and enhancing patient outcomes. The primary nursing method focuses on the therapeutic relationship with patients and families, creating the opportunity for nurses to develop their professional role where their technical and relational skills are equally valued and sustained [38]. The primary nursing method provides a high level of knowledge of the patient and his/her family, promoting in-depth case management. Primary nurses gradually become proficient in decision-making and continuity of care, evidencing advanced skills for the management of complex needs. Thus, nursing teams should have this profile of competencies and be prepared to apply this method to patients with advanced nursing care needs (e.g., patients who are expected to remain dependent on their caregivers after discharge, situations of transition from demanding care, serious chronic diseases, initial stage of the disease/dependence, sudden health transitions, or transitions that are difficult to accept).

Nonetheless, we believe that the work method to be adopted is essentially based on the way care is conceived and planned, its intentionality and awareness, and that it is later reflected in the way nurses organize, deliver, and evaluate this same care. Thus, it is not simply because a nurse collaborates with a shift colleague in a particular activity that we have to consider the functional nursing method or the task-oriented method. After all, if care is designed centered on the patient, it will necessarily and always give priority to the patient and not to the task.

## 5. Conclusions

Nursing is currently not regarded only as art but also as science in continuous (re)structuring, based on scientific principles and represented by its own body of knowledge. Because it is a developing science, with an object still searching for delimitation, the way to go includes working central concepts in theoretical models or paradigms that guide the knowledge of reality.

In all scientific and professional areas, the theory is expected to have repercussions in practice, structuring and contextualizing knowledge, circumscribing the reality under analysis, and predicting future events and needs. In nursing, the work methods are an example of the convergence between theory and practice, reflecting not only a set of socioeconomic and political impositions of their respective periods but also the successive paradigmatic transformations experienced. The primary nursing model dates to the years between 1890 and 1930 and is based on the principles described by Nightingale in her nursing notes. The functional nursing method emerged in the 1930s to 1940s, with the proliferation of clinical environments driven by the execution of tasks, bringing it closer to the scientific principles of management advocated by Taylor. Between 1950 and 1970, the infusion of humanist principles in health systems resulted in the adoption of the team nursing method, supported by Rogers’ principles of humanistic organizational psychology. Since 1980, the individual method, inspired by Orlando’s nursing process, emerged as a response to the socioeconomic pressures experienced in health organizations and combined with the increasing complexity of nursing care. 

Thus, when faced with the question “What will be the work method in my unit?”, a hybridization of methods that are commonly described as person-centered methods is observed, based on the key aspects mentioned. In short, this legitimate option that arises as a response to the different levels of complexity of care to be provided in different and differentiated work contexts requires making use of the advantages of each method. The implementation of a method in the clinical context is not expected to comply fully with all the theoretical requirements recommended given the following multifactorial barriers: (i) the conceptual framework adopted and the manager/leader’s conceptions of care; (ii) the qualifications and skills of the team nurses; (iii) the type of nursing care provided; (iv) the material and structural resources; (v) the units’ architectural characteristics; (vi) the excellence of care required; (vii) patient safety; (viii) the continuity of care; and (ix) the involvement and satisfaction of patients and families.

Likewise, the volume of the literature on work methods for nursing care delivery is vast and growing, especially when focused on processes (e.g., evidenced-informed nursing practice), pathways/approaches (e.g., Telford’s professional judgment; benchmarking; patient-nurse ratios), and tools (e.g., SBAR—Situation, Background, Assessment, Recommendation; Oulu Patient Classification) that can be used to select a method and comply with its main assumptions. However, there is no substantial evidence based on which process, pathway/approach, or tool should be selected. This is not necessarily a problem, but rather an adaptation to the needs because what distinguishes them is how care itself is conceptualized, rather than the activities that are developed.

## Figures and Tables

**Figure 1 ijerph-18-02088-f001:**
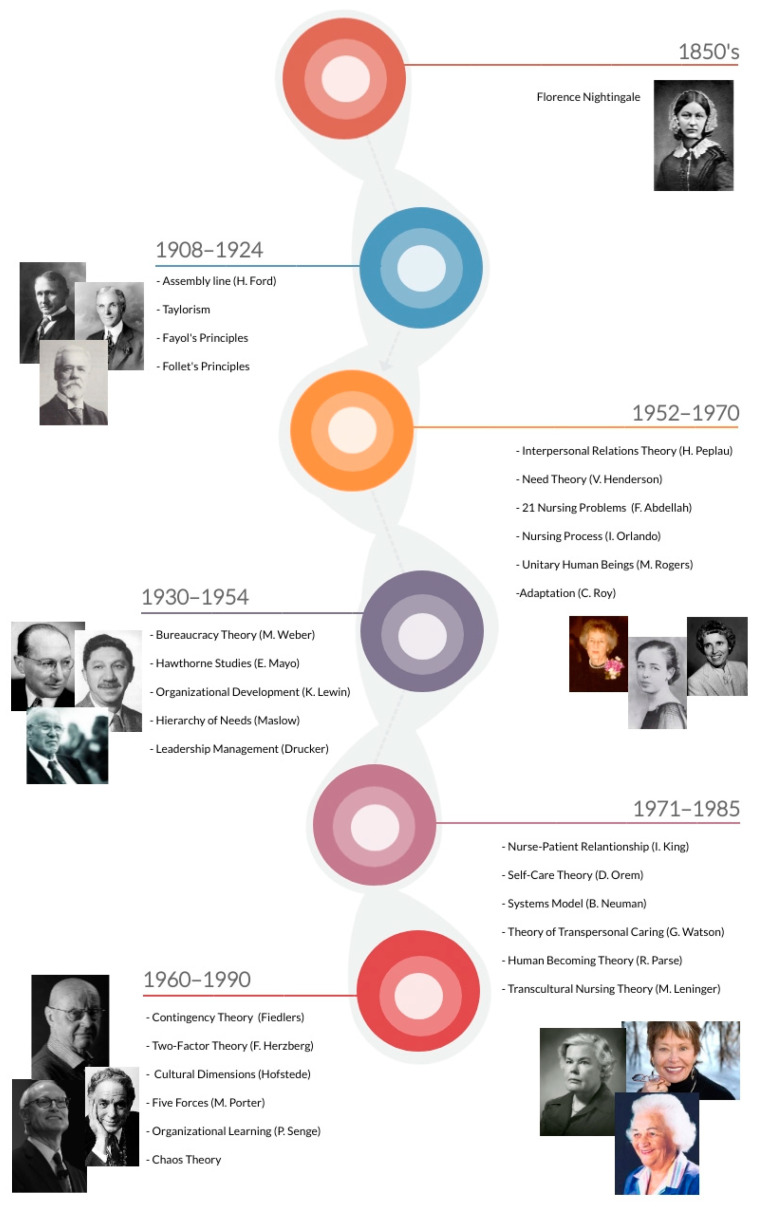
A historical perspective of management and nursing theories.

**Figure 2 ijerph-18-02088-f002:**
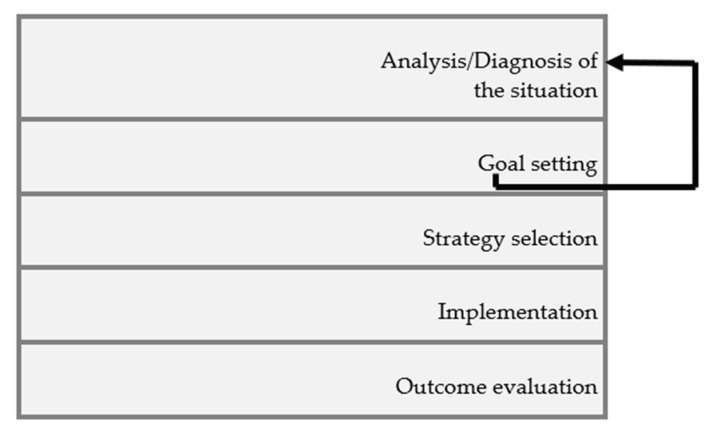
Phases of the process for planning the care delivery work method.

**Figure 3 ijerph-18-02088-f003:**
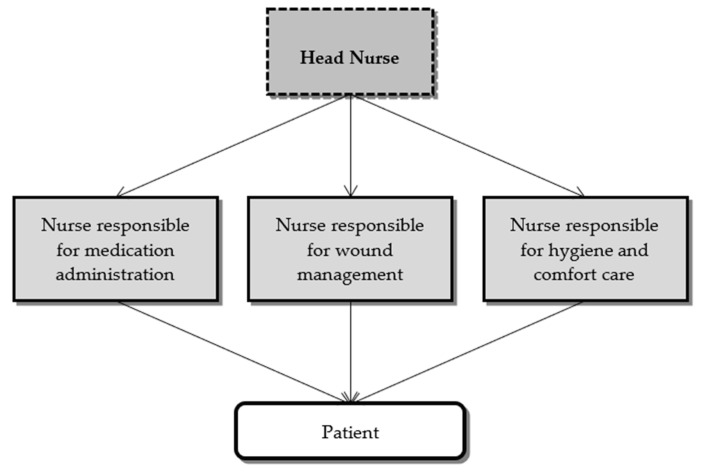
Structure of the functional nursing method.

**Figure 4 ijerph-18-02088-f004:**
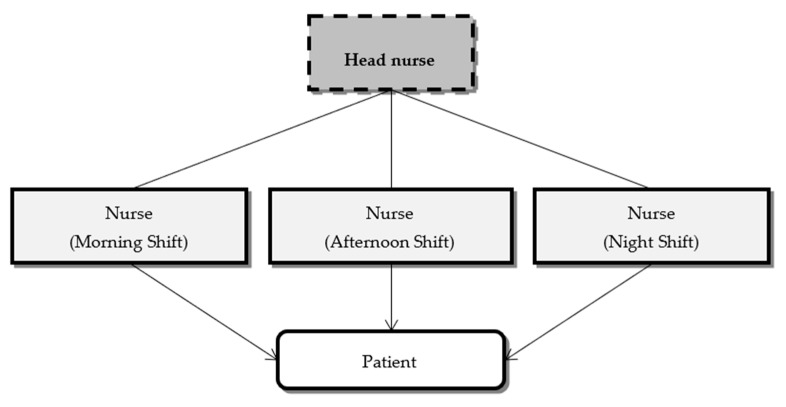
Structure of the individual method.

**Figure 5 ijerph-18-02088-f005:**
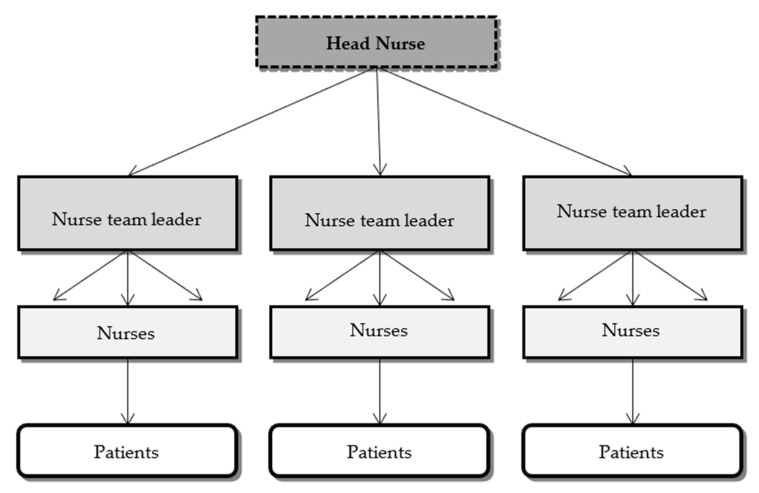
Structure of the team nursing method.

**Figure 6 ijerph-18-02088-f006:**
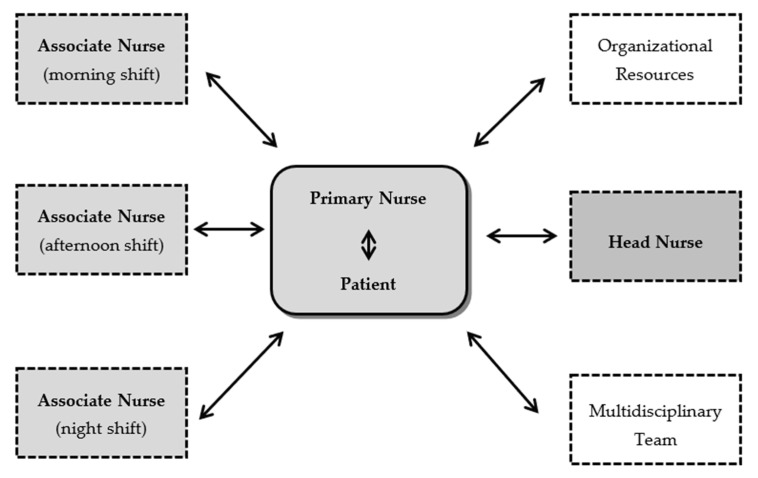
Structure of the Primary Nursing Method.

**Figure 7 ijerph-18-02088-f007:**
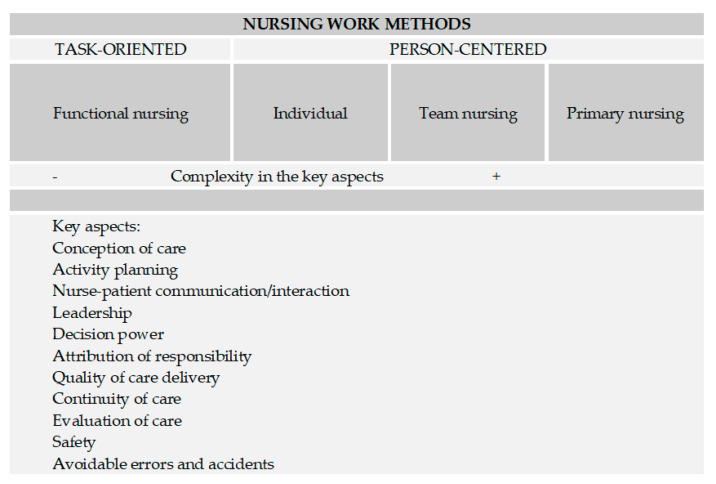
Key aspects of the methods of organization and delivery of nursing care.

**Table 1 ijerph-18-02088-t001:** Work methods in nursing.

Task-Focused	Person-Centered
Functional nursing	Individual
	Team nursing
	Primary nursing

## Data Availability

Not applicable.

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
