# Peer review of "Work Methods for Nursing Care Delivery"

_ijerph, 2021, doi:10.3390/ijerph18042088_

Round 1
Reviewer 1 Report
The auhtors have throuroughly conducted an analysis concerning the methods for Nursing care delivery. This is a very important and crucial subject in medicine. there is little attention for this subject in the literature, making this article very interesting and worth publishing.
This was a narative analysis, reporting an analysing the work methods for nursing care delivery. Managements theories have been explained and the course in time with the different concept in relation to time.
the article summarizes the concepts over time, with the different organization concepts. this is excellent and hlps organization in ocnetpion development and the chpise of organizing care.
The article was a bit too long, making it less atractive to read, alhtough the grammar was excellent.
The authors could considered summarizing the articel a bit, wihtout lossing the essential message of the article, to improve the attractiveness.
Author Response
We appreciate the comments made by Reviewer 1 and are grateful for its insightful analysis of our work. We have made some adjustments to the article following other commentary provided by different reviewers. We invite Reviewer 1 to read the new version of the manuscript and are open to any further suggestions that may be considered.
Reviewer 2 Report
Thank you for the opportunity to review this scholarly overview of significant conceptual frameworks for nursing practice. As the authors attest, "Nursing knowledge is the knowledge of synthesis that can be revealed only by accessing the practice" (lines 178-179). The authors contribute evidence-based validation of patient-centered nursing frameworks (Islam & Muhamad, 2021; Nyatuka & De La Harpe, 2021; Sentell et al., 2021) to the existing literature.
Thought leaders (Heinlein, 1958; Wilson, 1998) suggest synthesizers "put pieces together" (Heinlein, 1958, loc. 815) and are "able to put together the right information at the right time, think critically about it, and make important choices wisely" (Wilson, 1998, p. 294). In the same spirit, the authors utilize synthesis (lines 141, 160, 179, 201, 207, 208) to promote "hybridization of methods that are commonly described as person-centered" (line 576).
Opportunities for improvement:
Lines 54-57: Lengthy sentence; recommend shorten to improve readability.
Lines 68-69: "In this line of thought, which reflects the disruptive conceptions that characterized this last century . . ."; recommend identifying "disruptive conceptions" for the purposes of this manuscript.
Line 555: Expand "Final considerations" to include discussion of situational components of nursing practice. The "hybridization of methods" (line 575) option "arises as a response to the different levels of complexity of care to be provided in different and differentiated work contexts [aka situations]" (lines 577-578). The Joint Commission, Agency for Healthcare Research and Quality (AHRQ), Institute for Health Care Improvement (IHI), and World Health Organization (WHO) endorse SBAR (Situation, Background, Assessment, Recommendation) as an effective tool to promote patient safety (Shahid & Thomas, 2018). Situational awareness is vital to patient-centered nursing practice.
References:
Heinlein, R. A. (1958). Have space suit - will travel. Kindle Edition.
Islam, S., & Muhamad, N. (2021). Patient-centered communication: an extension of the HCAHPS survey. Benchmarking: An International Journal.
Nyatuka, D. R., & De La Harpe, R. (2021). Design considerations for patient‐centered eHealth interventions in an underserved context: A case of health and wellbeing services within Nairobi's informal settlements in Kenya. The Electronic Journal of Information Systems in Developing Countries, e12164.
Sentell, T., Foss-Durant, A., Patil, U., Taira, D., Paasche-Orlow, M. K., & Trinacty, C. M. (2021). Organizational health literacy: Opportunities for patient-centered care in the wake of COVID-19. Quality Management in Healthcare, 30(1), 49-60.
Shahid, S., & Thomas, S. (2018). Situation, background, assessment, recommendation (SBAR) communication tool for handoff in health care–a narrative review. Safety in Health, 4(1), 1-9.
Wilson, E. O. (1998). Consilience: The unity of knowledge. Random House.
Author Response
The authors would like to thank Reviewer 2 for the careful and insightful analysis of the manuscript. The opportunities for improvement identified by Reviewer 2 were formally address and we provide feedback below.
Reviewer 2: Lines 54-57: Lengthy sentence; recommend shorten to improve readability.
Authors: Thank you, we agree with this suggestion and shortened the sentence. Please see lines 54-62 of the manuscript.
Reviewer 2: Lines 68-69: "In this line of thought, which reflects the disruptive conceptions that characterized this last century . . ."; recommend identifying "disruptive conceptions" for the purposes of this manuscript.
Authors: Thank you for this suggestion. Although the main conceptions were identified in Figure 1, we have now included a brief description in the manuscript text. Please see line 68 onwards.
Line 555: Expand "Final considerations" to include discussion of situational components of nursing practice. The "hybridization of methods" (line 575) option "arises as a response to the different levels of complexity of care to be provided in different and differentiated work contexts [aka situations]" (lines 577-578). The Joint Commission, Agency for Healthcare Research and Quality (AHRQ), Institute for Health Care Improvement (IHI), and World Health Organization (WHO) endorse SBAR (Situation, Background, Assessment, Recommendation) as an effective tool to promote patient safety (Shahid & Thomas, 2018). Situational awareness is vital to patient-centered nursing practice.
Authors: We firmly agree with the Reviewer’s 2 suggestion of focusing on SBAR as a strategic pathway for clear communication between healthcare professionals. In fact, some of the authors of this manuscript have developed previous research in using SBAR in different clinical settings with nursing team.
Without compromising other reviewers’ suggestion of summarizing the article to reduce its current length, we have included a description of current processes, pathways/approaches, and tools (including the SBAR) described in current nursing literature as options that can assist nursing managers in the selection, implementation and assessment of a working method.
Reviewer 3 Report
This interesting article analyzes the work methods based on care design, identification of needs, care organization, planning, delivery, evaluation, continuity, safety, and complexity of care, and discharge preparation. In overall, the manuscript clearly responses to the aim of this work. It is very interesting for the readers' issue based on nursing theories. The authors had better develop in the text more extensively the phases of the process for planning the care delivery work method.
Author Response
We thank Reviewer 3 for his insightful analysis of our work and overall positive feedback on the manuscript.
The manuscript was submitted to slight revisions in accordance with other reviewers' commentary. We invite Reviewer 3 to read the new version and are open to any other suggestions that may arise.
Round 2
Reviewer 3 Report
The new version of the manuscript has been revised according to the comments and I have nothing to add as a suggestion. It would be a very interesting manuscript for the readers.